# Peripherally Inserted Central Venous Catheter (PICC) Related Bloodstream Infection in Cancer Patients Treated with Chemotherapy Compared with Noncancer Patients: A Propensity-Score-Matched Analysis

**DOI:** 10.3390/cancers15123253

**Published:** 2023-06-20

**Authors:** Romaric Larcher, Koko Barrigah-Benissan, Jerome Ory, Claire Simon, Jean-Paul Beregi, Jean-Philippe Lavigne, Albert Sotto

**Affiliations:** 1PhyMedExp (Physiology and Experimental Medicine), INSERM (French Institute of Health and Medical Research), CNRS (French National Centre for Scientific Research), University of Montpellier, 34000 Montpellier, France; 2Infectious and Tropical Diseases Department, Nimes University Hospital, Place du Professeur Robert Debre, 30000 Nimes, France; albert.sotto@chu-nimes.fr; 3VBIC (Bacterial Virulence and Chronic Infections), INSERM U1047, University of Montpellier, 34000 Montpellier, France; epiphanie08@yahoo.fr (K.B.-B.); jerome.ory@chu-nimes.fr (J.O.); jean.philippe.lavigne@chu-nimes.fr (J.-P.L.); 4Department of Microbiology and Hospital Hygiene, Nimes University Hospital, Place du Professeur Robert Debre, 30000 Nimes, France; 5Department of Pharmacy, Nimes University Hospital, Place du Professeur Robert Debre, 30000 Nimes, France; claire.simon@chu-nimes.fr; 6Department of Medical Imaging, Nimes University Hospital, Place du Professeur Robert Debre, 30000 Nimes, France; jean.paul.beregi@chu-nimes.fr

**Keywords:** PICC-line, CR-BSI, CRI, Gram-negative bacteria, catheter lumen, hematological malignancies, solid tumor

## Abstract

**Simple Summary:**

The use of peripherally inserted central catheters (PICCs) or PICC lines has become an established part of daily practice due to their ease of insertion, maintenance, and removal. Their use has increased particularly in cancer patients treated with chemotherapy who are immunocompromised and, therefore, known to have an increased risk of infection. However, the risk of PICC-related infections in this population compared to noncancer patients remains poorly evaluated. We found that the PICC-related bloodstream infection rate was more than twice as high in cancer patients compared to noncancer patients. In addition, we confirmed that dual-lumen PICCs had a higher risk of PICC-related bloodstream infection than single-lumen PICCs. Our results encourage physicians to carefully implement infection-control measures in cancer patients receiving chemotherapy through a PICC and particularly to limit the number of catheter lumens in these patients.

**Abstract:**

The use of peripherally inserted central catheters (PICCs) has increased in cancer patients. This study aimed to compare the incidence of PICC-related bloodstream infections (PICCR-BSIs) in cancer patients treated with chemotherapy and in noncancer patients. We performed a secondary analysis from a retrospective, single-center, observational cohort. The PICCR-BSI incidence rates in cancer and noncancer patients were compared after 1:1 propensity-score matching. Then, the factors associated with PICCR-BSI were assessed in a Cox model. Among the 721 PICCs (627 patients) included in the analysis, 240 were placed in cancer patients for chemotherapy and 481 in noncancer patients. After propensity-score matching, the PICCR-BSI incidence rate was 2.6 per 1000 catheter days in cancer patients and 1.0 per 1000 catheter days in noncancer patients (*p* < 0.05). However, after adjusting for variables resulting in an imbalance between groups after propensity-score matching, only the number of PICC lumens was independently associated with PICCR-BSI (adjusted hazard ratio 1.81, 95% confidence interval: 1.01–3.22; *p* = 0.04). In conclusion, the incidence rate of PICCR-BSI is higher in cancer patients treated with chemotherapy than in noncancer patients, but our results also highlight the importance of limiting the number of PICC lumens in such patients.

## 1. Introduction

Peripherally inserted central catheters (PICCs), also known as PICC lines, are venous catheters inserted into a peripheral vein in the upper arm, the distal tip of which is located in the territory of the superior vena cava [1,2]. They are easier to place and less prone to complications at the time of insertion than other central venous catheters (CVC) [1,2]. PICCs are mainly used for the administration of parenteral nutrition, prolonged antimicrobial therapy, or chemotherapy [1,2]. In some patients requiring simultaneous administration of these drugs, devices with two lumens (two separate tubings in the same catheter) can be inserted [3]. The different types of PICCs can be used for durations of up to 6 months and even longer [1,2].

Over the past decade, the use of PICCs has become more widespread in daily practice, leading at the same time to an increase in the number of PICC-related complications [1,4]. Complications include mechanical complications such as catheter dysfunction or accidental removal [5], thrombotic complications [6], and infectious complications [1,7]. PICC-related infections (PICCRIs) and, in particular, PICC-related bloodstream infections (PICCR-BSIs) are the most frequent complications [1,7]. However, PICCs would have a potential advantage over other central venous catheters since they would reduce the risk of catheter-related bloodstream infections compared to other CVCs [8,9,10].

In cancer patients treated with chemotherapy, the use of PICCs has increased dramatically [11] but studies provide conflicting data regarding the risk of CVC-related complications, with some reporting that implantable port catheters are safer [12], others that PICCs are safer [13], and finally, some authors report that both devices are equivalent [14]. Moreover, the risk of PICC-related complications in cancer patients compared with noncancer patients remains poorly assessed [15]. The complication rate appears to be higher in cancer patients than in noncancer patients, especially with regard to the PICC thrombosis rate [16,17]. In the literature, the incidence rates of PICCR-BSI in cancer patients range between 2 and 4 per 1000 catheter days [18,19,20,21,22] and between 1 and 2 per 1000 catheter days in noncancer patients [23,24], but few studies have compared the incidence rate of PICCR-BSI in cancer patients treated with chemotherapy and in noncancer patients.

We, therefore, conducted a study that aimed to compare the incidence rate of PICCR-BSI in cancer patients treated with chemotherapy and in noncancer patients and to describe the microorganisms involved in these patients.

## 2. Materials and Methods

### 2.1. Study Design and Settings

We performed a secondary analysis from a retrospective, single-center, observational cohort [1] that included consecutive adult inpatients and outpatients who had at least one PICC insertion at Nimes University Hospital from 1 April 2018 to 1 April 2019.

In this 2094-bed teaching hospital, ultrasound-guided insertions of single- or double-lumen PICC (Bard Access Systems, Salt Lake City, UT, USA) were performed 5 days per week in the medical imaging department under aseptic conditions according to the French Society of Infection Control (SF2H) guidelines [25]. The position of the PICC was then checked by chest X-ray and adjusted if necessary, and saline was used to avoid occlusion of the lumen(s) [25]. All PICCs were inserted by a trained radiologist or radiology technician in an interventional radiology room.

### 2.2. Patients

All patients of the Barrigah-Benissan et al. cohort [1] were screened. Only cancer patients treated with chemotherapy and noncancer patients were included in the current study. Cancer patients were defined as patients undergoing treatment for a solid tumor with or without metastasis or for a hematological malignancy (leukemia, lymphoma, or myeloma). Cancer patients not treated with chemotherapy (treatment completed or in palliative care) were excluded.

### 2.3. Data Collection

The following variables were collected: age, sex, body mass index (BMI), Charlson comorbidity index, type of cancer, ongoing immunosuppressive treatment (corticosteroids or other immunosuppressive drugs), the reason for PICC insertion, number of lumens, side of PICC insertion site, the reason for PICC removal, PICC duration, and vital status at PICC removal. The occurrence and timing of PICC colonization or PICCRI were also collected. The bacteria species involved in PICC colonization and PICCRI were recorded. 

To secure the diagnosis of PICCRI [26], all medical records were reviewed by an adjudication committee composed of an intensivist, an infection control specialist, and an infectious disease physician. If there was a discrepancy, the PICCRI diagnosis was discussed among the committee members until a consensus was achieved.

### 2.4. Microbiology

The department of microbiology performed quantitative culture of the distal segment of intravascular catheters as described by Brun-Buisson. BD BACTEC^™^ Aerobic Plus and BD BACTEC^™^ Anaerobic Plus blood-culture bottles were placed in the BD BACTEC^™^ FX system (Becton-Dickinson, Franklin Lakes, NJ, USA) for incubation up to 5 and 7 days, respectively, or until automatic detection of positivity. In cases of suspected endocarditis, the total incubation period was 14 days. If the bottle was not positive during the incubation period, it was considered negative. For bottles that detected positive, the detection time was recorded; then, Gram strain and subculture for incubation for 24 h at 35 °C were performed. Bacterial and fungal identification were performed using mass spectrometry Vitek^®^ MS (bioMérieux, Marcy-l’Etoile, France) and antimicrobial susceptibility testing (AST) using disk diffusion method on Mueller–Hinton agar (Bio-Rad, Hercules, CA, USA), according to the European committee on antimicrobial susceptibility testing (EUCAST) guidelines [27].

### 2.5. Study Definitions

PICC colonization were defined as a quantitative culture ≥10^3^ CFU/mL, according to Brun-Buisson, without bacteremia or clinical signs [26]. 

No-bacteremia PICCRI (NB-PICCRI), were defined, in the absence of bacteremia, as a combination of (i) PICC culture ≥10^3^ CFU/mL and (ii) (a) signs of local infection (purulent discharge from the PICC insertion site); and/or (b) systemic signs, with complete or partial resolution of systemic signs of infection within 48 h after PICC removal [26].

PICCR-BSI were defined as an association of (i) the occurrence of either bacteremia or fungaemia during the 48-h period surrounding PICC removal (or a suspected diagnosis of PICCRI when the PICC is not removed immediately); (ii) and either a positive culture with the same microorganism on one of the following samples: insertion site culture, or PICC culture ≥10^3^ CFU/mL or positive central and peripheral blood cultures with the same microorganism, with a central/peripheral positive blood culture time lag > 2 h with central blood cultures being positive earlier than the peripheral ones [26].

PICCRI were defined as NB- PICCRI and PICCR-BSI [26].

### 2.6. Statistical Analysis

PICC insertion was the unit for statistical analyses. The categorical data were described as numbers and percentages, and continuous data as medians with 25th and 75th percentiles (interquartile range: IQR). Patients were segregated according to cancer (yes or no). The categorical variables were compared by Chi-square or Fisher’s exact test, and the continuous variables were compared by Student’s *t* test or Wilcoxon’s rank-sum test, as appropriate. 

Propensity-score matching was performed to compare the incidence of PICCR-BSI in cancer patients treated with chemotherapy with those in noncancer patients. Patients were matched (1:1) with the algorithm for nearest-neighbor matching without replacement, using a maximum tolerance distance between the matched subjects of 0.1 standard deviation. The confounding variables used to calculate the propensity scores were age, BMI, number of PICC lumens, and Charlson comorbidity index. 

We performed survival analyses to consider the time dimension. The observation time was the time from PICC insertion to the occurrence of the event (PICCR-BSI) and/or PICC removal. We identified the variables resulting in an imbalance between groups after propensity score matching by calculating the standardized mean difference; then, we included them in the subsequent Cox proportional hazards model as covariates to assess the effect of chemotherapy-treated cancer on the incidence of PICCR-BSI. Cumulative incidence curves of PICCR-BSI were obtained by the Kaplan–Meier methodology and compared using the log-rank test.

All tests were two-sided, and a *p*-value of less than 0.05 was considered statistically significant. Analyses were performed using the R software version 4.2.2 (The R Foundation for Statistical Computing, Vienna, Austria).

## 3. Results

### 3.1. Patients and Peripherally Inserted Central Venous Catheters

Of the 901 PICCs inserted in 783 patients in the initial cohort [1], 721 inserted PICCs in 627 patients were included in the analysis corresponding to 31,831 catheter days. Among the PICCs included, 240 were placed in cancer patients for chemotherapy and 481 in noncancer patients, corresponding to 15,108 and 16,723 catheter days, respectively (Appendix A). The median age of the study population was 69 years (IQR: 57, 79) and 55% of PICCs were inserted in male patients. Two thirds of the cancer patients had a solid tumor and one third had hematological malignancies (Table 1).

Cancer patients were younger, had lower BMIs, and higher Charlson score than noncancer patients. Double-lumen PICCs were more frequently placed in cancer patients who also have longer PICC indwelling time than noncancer patients, 32 days (IQR: 15, 76) versus 17 days (IQR: 8, 35).

In noncancer patients, PICCs were removed primarily at the end of treatment because they were no longer useful (70%). On the contrary, in more than half of cancer patients (57%), PICCs were removed due to a suspected or confirmed complication.

### 3.2. Incidence of PICC-Related Complications

The incidence of PICC-related complications was similar between cancer and noncancer patients, except for PICCR-BSI (Table 2).

The incidence of PICCR-BSI was 2.6 per 1000 catheter days in cancer patients and 1.1 per 1000 catheter days in noncancer patients (*p* = 0.07), see Figure 1.

### 3.3. Microbiology

Cancer patients mainly had PICCR-BSI caused by Gram-negative bacteria, especially Enterobacterales, and nonfermenters, whereas noncancer patients more frequently had PICCR-BSI caused by Gram-positive bacteria, mainly coagulase-negative staphylococci, and PICC-related fungemia. It should be noted that no cancer patients had PICC-related fungemia and noncancer patients did not have PICCR-BSI caused by nonfermenters. 

Bacterial species involved in PICCR-BSI are presented in Table 3 (see also Appendix A).

### 3.4. Incidence of PICC-Related Complications after Propensity Score Matching

After propensity-score matching on age, BMI, number of PICC lumens, and Charlson comorbidity index, the rate of PICCR-BSI remains higher in cancer patients than in noncancer patients at 17% and 2.9%, respectively (*p* < 0.001), corresponding to a PICCR-BSI incidence rate of 2.6 per 1000 catheter days in cancer patients treated with chemotherapy and 1 per 1000 catheter days in noncancer patients (Table 4).

Despite matching, the population remained unbalanced in terms of age and number of PICC lumens. After adjustment on these confounders, the risk for PICCR-BSI in patients treated with chemotherapy for cancer remained higher, but the difference was no longer statistically significant: adjusted hazard ratio (aHR) 1.83 confidence interval at 95% (95%CI): 0.8699–3.858 (*p* = 0.11). On the contrary, double-lumen PICC placement was independently associated with an increased incidence of PICCR-BSI, aHR 1.81, 95%CI: 1.01–3.22 (*p* = 0.04), see Table 5.

## 4. Discussion

We reported herein the results of a large cohort of 721 PICC placements in 627 patients (31,831 catheter days) showing a PICCR-BSI incidence rate of 1.8 per 1000 catheter days. In cancer patients, the PICCR-BSI incidence rate was 2.6 per 1000 catheter days and PICCR-BSIs were mainly caused by Gram-negative bacteria, whereas in noncancer patients, the PICCR-BSI incidence rate was 1.1 per 1000 catheter days and PICCR-BSIs were mostly caused by Gram-positive bacteria. After propensity-score matching, the PICCR-BSI incidence rate remained more than twofold higher in cancer patients (2.6 versus 1 per 1000 catheter days). However, after adjusting the variables resulting in an imbalance between groups after propensity score matching, only the number of PICC lumens was independently associated with PICCR-BSI.

The incidence rates of PICCR-BSI reported in the literature range widely, from 1.0 to 2.1 per 1000 catheter days in noncancer patients [23,24] and from 2.0 to 4.0 per 1000 catheter days in patients with hematological malignancies [20,21,22] or solid tumors [8]. We found that the incidence rate of PICCR-BSI was twice as high in cancer patients treated with chemotherapy as in noncancer patients, which is consistent with previous reports [8,20,21,22,23,24]. Among cancer patients, those with hematological malignancies, especially those with leukemia or high-grade lymphoma, are at higher risk for PICCRI compared with patients with solid tumors [11]. In addition, neutropenic patients with bloodstream infections are at higher risk of mortality compared with nonneutropenic patients [11].

Our results confirm that hematological malignancies and solid cancers with ongoing chemotherapy are risk factors associated with PICCR-BSI [3]. However, they also highlighted the importance of limiting the number of PICC lumens to the minimum required. The use of multilumen PICCs or the presence of another central venous catheter at the time of PICC placement have already been reported as risk factors for PICCR-BSI [3,28]. The cancer patient remains fragile and susceptible to infections; therefore, maximum precautions should be taken to limit PICCRI. It seems also important to limit the catheter dwell time [9], particularly in patients with hematological malignancies [20]. Although this risk factor remains debated [22], many studies [1,20,29] encouraged clinicians to limit the PICC indwelling time to approximately 4 weeks. Especially since the risk of PICCR-BSI does not seem to increase with multiple PICC insertions [20]. Moreover, our results confirmed that the PICCR-BSI rate was not influenced by the side of the PICC insertion site [21]. In addition to limiting the number of PICC lumens and catheter dwell time, it is necessary to perform dressing changes every 4 to 7 days in aseptic conditions, to disinfect the administration sites at each use, to improve hand-hygiene compliance, and to monitor clinical signs of infection [25]. Some authors have also reported the interest of self-management to reduce complications in cancer patients receiving chemotherapy through a PICC [30]. In particular, the importance of self-monitoring for clinical signs of infection should be emphasized [1]. 

As the use of PICCs is booming, some authors have suggested placing PICCs at the patient’s bedside to facilitate access to these devices [31,32]. This practice does not seem to increase PICC-related complications, including infections, but studies [31,32] are mainly carried out in intensive care units and include few or no cancer patients. As cancer patients have a higher risk for PICCR-BSI, a careful evaluation of this type of practice will be required before it becomes routine in the oncology and hematology departments. Importantly, no PICCs were inserted at the bedside in our cohort. Thus, our results encourage physicians in charge of cancer patients treated with chemotherapy to favor PICC placement in an interventional radiology room that is a controlled environment with air filtration systems capable of delivering clean, filtered, and contaminant-free air into the room, and in which biocleaning is performed between each PICC placement. To prevent PICCRI, a trained operator must observe appropriate infection-control measures, such as hand hygiene, skin preparation with an alcohol antiseptic, maximal sterile-barrier precautions, and an aseptic technique during PICC placement [25]. In addition to the particular attention that must be paid to limiting catheter dwelling time and the number of lumens, patients and caregivers must be educated to apply strict hand hygiene and skin antisepsis during PICC care and dressing management and to detect and recognize signs of PICCRI and other PICC-related complications [1].

Another striking finding of our study highlighting the peculiarities of cancer patients is that Gram-negative bacteria are an increasing cause of PICCR-BSI in this population. Over the last decade, Gram-negative bacteria have become the main etiological microorganisms of catheter-related bloodstream infections [33,34,35]. The immunocompromised patients are at the greatest risk of being infected by their own enterobacteria [36]; accordingly, more than half of the bacteria involved in PICCR-BSI of cancer patients are Gram-negative bacilli [34,35]. In contrast, Gram-positive bacteria remained responsible for most catheter-related infections in noncancer patients, followed by gram-negative bacteria and fungi [24,28], although a change in this trend has been suggested [37,38,39].

This study has limitations. First, its single-center design could limit the extrapolation of the results, as PICC-related complication rates vary between hospitals [15]. However, our cohort included a large number of patients from four different oncology departments and one hematology department, representing a mix of different practices. In addition, this study is one of the few to assess the risk of PICCR-BSI in cancer and noncancer patients in a recent period. Second, we studied only one type of device, whereas the rate of complications differed across the PICC types [18]. Nonetheless, the rate of PICCR-BSI did not [18]. Third, the retrospective design of the study limits our analyses to the available data in medical records and may induce bias in data collection and results interpretation. Some risk factors such as the receipt of total parenteral nutrition through the PICC [3], the neutrophil count at the time of PICC placement or infection [11], or the degree of dependence of patients could not be assessed, nor could the outpatient/inpatient status. However, the data suggested that using PICCs in outpatients is not associated with an elevated risk of complications [40], including in cancer patients [41].

## 5. Conclusions

In a large French retrospective cohort study, we found that the incidence rate of PICCR-BSI was 2.6 per 1000 patient days in cancer patients treated with chemotherapy and 1.1 per 1000 patient days in noncancer patients. We showed that PICCR-BSIs were most often caused by Gram-negative bacteria in cancer patients whereas they were mainly caused by Gram-positive bacteria in noncancer patients. The incidence rate of PICCR-BSI remained higher in cancer patients treated with chemotherapy than in noncancer patients after propensity-score matching. However, our results suggest that the use of double-lumen PICCs in cancer patients may be a higher risk of PICCR-BSI than the immunosuppression induced by cancer treatment with chemotherapy.

Further multicenter studies are mandatory to better understand the reasons for the increase in PICCRI caused by Gram-negative bacteria, to assess the risk of PICCR-BSI in cancer patients, and to determine whether measures to prevent PICCRI, such as limiting the number of PICC lumens, improve outcomes in these patients.

## Figures and Tables

**Figure 1 cancers-15-03253-f001:**
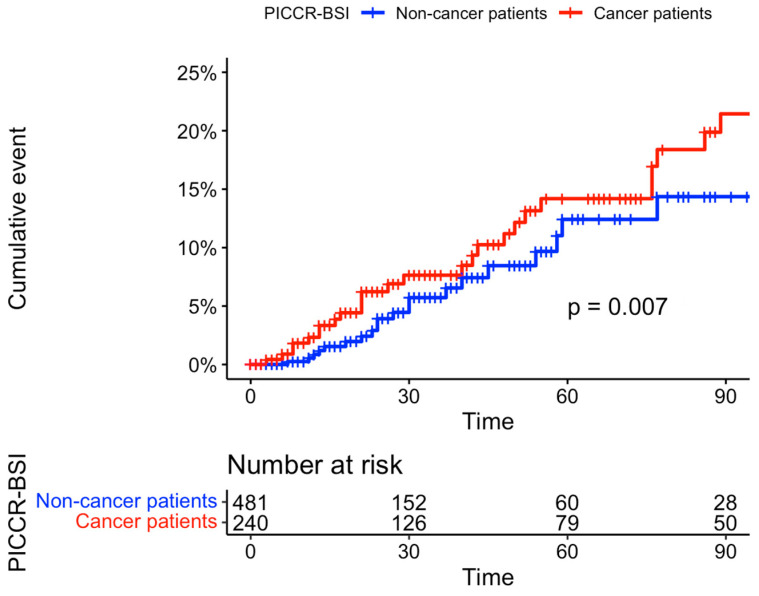
Cumulative incidence curves of peripherally inserted central catheter-related bloodstream infection (PICCR-BSI) in cancer (red) and noncancer patients (blue).

**Table 1 cancers-15-03253-t001:** Characteristics of the study population and peripherally inserted central catheters (PICC).

Characteristics	Overall *n* = 721 ^1^	Cancer Patients *n* = 240 ^1^	Noncancer Patients *n* = 481 ^1^	*p*-Value ^2^
Demographics:				
Age	69 (57, 79)	66 (54, 74)	72 (60, 82)	<0.001
Male	399 (55%)	129 (54%)	270 (56%)	0.5
Body Mass Index (BMI)	24 (21, 29)	23 (21, 27)	25 (21, 30)	<0.001
Charlson comorbidity index	6 (3, 8)	6 (4, 9)	5 (3, 7)	<0.001
Cancer type:				
Solid tumor	161 (22%)	161 (67%)	-	
Localized solid tumor	55 (7.6%)	55 (23%)	-	
Metastatic solid tumor	106 (15%)	106 (44%)	-	
Hematological malignancies	79 (11%)	79 (33%)	-	
Leukemia	43 (6.0%)	43 (18%)	-	
Lymphoma	18 (2.5%)	18 (7.5%)	-	
Myeloma	18 (2.5%)	18 (7.5%)	-	
Main reason for PICC placement:				
Cancer chemotherapy	240 (33%)	240 (100%)	-	
Antimicrobial therapy	306 (42%)	-	306 (64%)	
Limited peripheral access	109 (15%)	-	109 (23%)	
Long-term venous access	31 (4.3%)	-	31 (6.4%)	
Parenteral nutrition	35 (4.9%)	-	35 (7.3%)	
Double-lumen PICC	155 (21%)	104 (43%)	51 (11%)	<0.001
Right side PICC insertion site	167 (23%)	51 (21%)	116 (24%)	0.4
Reason for PICC removal:				
End of intravenous therapy	426 (59%)	89 (37%)	337 (70%)	<0.001
Port implantation	23 (3.2%)	18 (7.5%)	5 (1.0%)	<0.001
Mechanical complication	67 (9.3%)	20 (9.8%)	47 (9.8%)	>0.9
PICCRI ^3^ (suspected or confirmed)	123 (17%)	77 (32%)	46 (9.6%)	<0.001
Death	82 (11%)	36 (15%)	46 (9.6%)	0.03
PICC duration (days)	21 (10, 46)	32 (15, 76)	17 (8, 35)	<0.001
Number of catheter days	31,831	15,108	16,723	

^1^ Median (interquartile range) or *n* (%); ^2^ Wilcoxon rank sum test; Pearson’s Chi-squared test; Fisher’s exact test as appropriate; ^3^ PICCRI: PICC-related infection.

**Table 2 cancers-15-03253-t002:** Peripherally inserted central catheter (PICC) related complications (rates and incidences).

PICC Related Complications	Overall ^1^ *n* = 721	Cancer Patients ^1^ *n* = 240	Noncancer Patients ^1^ *n* = 481	*p*-Value ^2^
Accidental removal (rate)	47 (6.5%)	10 (4.2%)	37 (7.7%)	0.071
Accidental removal per 1000 catheter days	1.5	0.7	2.2	
Vein thrombosis (rate)	14 (1.9%)	8 (3.3%)	6 (1.2%)	0.082
Vein thrombosis per 1000 catheter days	0.4	0.5	0.4	
Catheter dysfunction (rate)	6 (0.8%)	2 (0.8%)	4 (0.8%)	>0.9
Catheter dysfunction per 1000 catheter days	0.2	0.1	0.2	
PICC colonization (rate)	33 (4.6%)	15 (6.2%)	18 (3.7%)	0.13
PICC colonization per 1000 catheter days	1.0	0.9	1.1	
NB-PICCRI ^3^ (rate)	11 (1.5%)	5 (2.1%)	6 (1.2%)	0.5
NB-PICCRI per 1000 catheter days	0.3	0.3	0.4	
PICCR-BSI ^4^ (rate)	58 (8.0%)	40 (17%)	18 (3.7%)	<0.001
PICCR-BSI per 1000 catheter days	1.8	2.6	1.1	

^1^ Median (IQR) or *n* (%); ^2^ Wilcoxon rank sum test; Pearson’s Chi-squared test; Fisher’s exact test as appropriate; ^3^ NB-PICCRI: nonbacteremia PICC-related infection; ^4^ PICCR-BSI: PICC-related bloodstream infection.

**Table 3 cancers-15-03253-t003:** Bacterial species involved in a peripherally inserted central catheter (PICC) related bloodstream infections.

Bacterial Species	Overall ^1^ *n* = 58	Cancer Patients ^1^ *n* = 40	Noncancer Patients ^1^ *n* = 18
Gram-negative bacteria	38 (66%)	31 (78%)	7 (39%)
Enterobacterales ^2^	29 (50%)	22 (55%) ^2^	7 (39%)
*Escherichia coli*	8 (14%)	5 (13%)	3 (17%)
*Enterobacter cloacae*	7 (12%)	6 (15%)	1 (6%)
*Klebsiella pneumoniae*	5 (9%)	4 (10%)	1 (6%)
*Klebsiella oxytoca*	3 (5%)	3 (8%)	0 (0%)
*Serratia marcescens*	2 (3%)	1 (3%)	1 (6%)
*Citrobacter koseri*	1 (2%)	0 (0%)	1 (6%)
*Hafnia alvei*	1 (2%)	1 (3%)	0 (0%)
*Klebsiella aerogenes*	1 (2%)	1 (3%)	0 (0%)
*Proteus mirabilis*	1 (2%)	1 (3%)	0 (0%)
Nonfermenters	9 (16%)	9 (23%)	0 (0%)
*Acinetobacter baumannii*	3 (5%)	3 (8%)	0 (0%)
*Acinetobacter ursingii*	1 (2%)	1 (3%)	0 (0%)
*Stenotrophomonas maltophilia*	2 (3%)	2 (5%)	0 (0%)
*Achromobacter xylosoxidans*	1 (2%)	1 (3%)	0 (0%)
*Pseudomonas aeruginosa*	1 (2%)	1 (3%)	0 (0%)
*Rhizobium radiobacter*	1 (2%)	1 (3%)	0 (0%)
Gram-positive bacteria	26 (45%)	17 (43%)	9 (50%)
*Staphylococcus epidermidis*	13 (22%)	7 (18%)	6 (30%)
*Staphylococcus aureus*	4 (7%)	4 (10%)	0 (0%)
*Staphylococcus haemolyticus*	1 (2%)	0 (0%)	1 (6%)
*Streptococcus pasteurianus*	3 (5%)	2 (5%)	1 (6%)
*Streptococcus mitis*	1 (2%)	1 (3%)	0 (0%)
*Enterococcus faecium*	2 (3%)	1 (3%)	1 (6%)
*Enterococcus faecalis*	1 (2%)	1 (3%)	0 (0%)
*Bacillus licheniformis*	1 (2%)	1 (3%)	0 (0%)
Fungi	4 (8%)	0 (0%)	4 (22%)
*Candida glabrata*	2 (3%)	0 (0%)	2 (11%)
*Candida parapsilosis*	2 (3%)	0 (0%)	2 (11%)

^1^ *n* (%); ^2^ including 10 AmpC beta-lactamase-producing Enterobacterales, 6 extended spectrum beta-lactamase-producing Enterobacterales and 5 carbapenem-resistant Enterobacterales.

**Table 4 cancers-15-03253-t004:** Characteristics of the study population and peripherally inserted central catheters (PICC) related complications (rates and incidences).

Characteristics	Overall *n* = 480 ^1^	Cancer Patients *n* = 240 ^1^	Noncancer Patients *n* = 240 ^1^	*p*-Value ^2^
Age	68 (54, 77)	66 (54, 74)	71 (54, 82)	0.003
Male	276 (57%)	129 (54%)	147 (61%)	0.10
Body Mass Index (BMI)	23 (20, 28)	23 (21, 27)	24 (20, 28)	0.9
Charlson comorbidity index	6 (3, 9)	6 (4, 9)	6 (3, 9)	0.12
Double-lumen PICC	154 (32%)	104 (43%)	50 (21%)	<0.001
PICC duration (days)	24 (11, 54)	32 (15, 76)	16 (8, 34)	<0.001
Number of catheter days	22,432	15,108	7324	
Accidental removal (rate)	27 (5.6%)	10 (4.2%)	17 (7.1%)	0.2
Accidental removal per 1000 catheter days	1.2	0.7	2.3	
Vein thrombosis (rate)	11 (2.3%)	8 (3.3%)	3 (1.3%)	0.13
Vein thrombosis per 1000 catheter days	0.5	0.5	0.4	
Catheter dysfunction (rate)	6 (1.3%)	2 (0.8%)	4 (1.7%)	0.7
Catheter dysfunction per 1000 catheter days	0.3	0.1	0.5	
PICC colonization ^3^ (rate)	23 (4.8%)	15 (6.2%)	8 (3.3%)	0.13
PICC colonization per 1000 catheter days	1.0	0.9	1.1	
NB-PICCRI ^4^ (rate)	10 (2.1%)	5 (2.1%)	5 (2.1%)	>0.9
NB-PICCRI per 1000 catheter days	0.4	0.3	0.7	
PICCR-BSI ^5^ (rate)	47 (9.8%)	40 (17%)	7 (2.9%)	<0.001
NB-PICCRI per 1000 catheter days	2.1	2.6	1.0	

^1^ Median (IQR) or *n* (%); ^2^ Wilcoxon rank sum test; Pearson’s Chi-squared test; Fisher’s exact test as appropriate; ^3^ PICCRI: PICC-related infection; ^4^ NB-PICCRI: nonbacteremia PICC-related infection; ^5^ PICCR-BSI: PICC-related bloodstream infection.

**Table 5 cancers-15-03253-t005:** Multivariable Cox model for peripherally inserted central catheters (PICC) related bloodstream infection after propensity score matching.

Variables	aHR ^1^	95%CI ^2^	*p*-Value
Cancer	1.83	0.86–3.86	0.11
Age (year)	1.003	0.98–1.02	0.75
Double-lumen PICC	1.81	1.01–3.22	0.04

^1^ aHR: adjusted hazard ratio; ^2^ 95%CI: confidence interval at 95%.

## Data Availability

The authors consent to share the collected data with others. The raw data supporting the conclusions of this article will be made available by the authors, without undue reservation. Data will be available immediately after the main publication and indefinitely.

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
