# Peer review of "Peripherally Inserted Central Venous Catheter (PICC) Related Bloodstream Infection in Cancer Patients Treated with Chemotherapy Compared with Noncancer Patients: A Propensity-Score-Matched Analysis"

_cancers, 2023, doi:10.3390/cancers15123253_

Round 1

Reviewer 1 Report

Presentation of methods and results are concise and succinct. Figures and Tables are well-organized and easy to interpret. Manuscript is well-written and addresses an interesting question that has clinical implications. The limitations of the study are addressed in the discussion. Conclusions are justified and supported by the data and analysis. It would be interesting to discuss how the PICC lines were placed and did it influence incidence of PICC line associated infections. PICC lines are sometimes placed at the bedside by trained staff or placed in interventional radiology suites by physicians.  It would also be interesting to see if the white blood cell count or absolute neutrophil count at time PICC line was placed had implications on incidence of PICC related blood stream infections. 

No significant issues with spelling or grammar for this manuscript. 

Author Response

Presentation of methods and results are concise and succinct. Figures and Tables are well-organized and easy to interpret. Manuscript is well-written and addresses an interesting question that has clinical implications. The limitations of the study are addressed in the discussion. Conclusions are justified and supported by the data and analysis.

First, we thank the reviewer for their interest to this work and for their remarks.

It would be interesting to discuss how the PICC lines were placed and did it influence incidence of PICC line associated infections. PICC lines are sometimes placed at the bedside by trained staff or placed in interventional radiology suites by physicians.

As the reviewer correctly pointed out, the room where the PICC is inserted can influence the infection rate, especially in cancer patients. Studies assessing the risk associated with bedside placement of PICC were performed mainly in ICU setting (Kim PlosOne 2019 and Yoon J. Pers. Med. 2023), include few cancer patients, and conclude that infection rates are similar to those after placement in interventional radiology. In our study, all PICCs were inserted in an interventional radiology room by a trained radiologist or radiology technician, and none are inserted at the patient's bedside. Thus, our study did not allow us to evaluate the insertion of PICCs at the patient's bedside, nor to compare infection rates between insertion at bedside and in the interventional radiology room. We have amended the discussion section in accordance with the reviewer's comment.

It would also be interesting to see if the white blood cell count or absolute neutrophil count at time PICC line was placed had implications on incidence of PICC related blood stream infections.

We completely agree with the reviewer, that it would be interesting to see if white blood cell count or neutrophil count at the time of PICC placement has an impact on the incidence of PICC-related bloodstream infections. However, this would require a dedicated study in which a non-neutropenic patient group is compared to a neutropenic patient group. Moreover, we have not designed our study in this way, so we do not have the data needed to compare such groups. On the other hand, as mentioned in the discussion, other studies have shown that neutropenic patients are at greater risk of PICCRI, but also of death linked to these infections. Please see discussion second paragraph and limit section.

Reviewer 2 Report

General comments:

In the manuscript entitled “Peripherally inserted central venous catheter (PICC) related bloodstream infection in cancer patients treated with chemotherapy compared with non-cancer patients: a propensity score-matched analysis”, the authors reveal that PICC-related bloodstream infection rate was higher in cancer patients than that of non-cancer patients. Moreover, the authors also demonstrate that PICC-related bloodstream infections were frequently caused by Gram-negative bacteria in cancer patients; meanwhile, they were mainly caused by Gram-positive bacteria in non-cancer patients. The manuscript is well-written and clear, and the data are presented logically.

Specific comment:

Interestingly, Gram-negative bacteria were frequently detected in PICC-related bloodstream infections developed in cancer patients. The authors are advised to briefly mention the microbial procedures to detect the causative pathogens in their hospital.

Author Response

General comments:

In the manuscript entitled “Peripherally inserted central venous catheter (PICC) related bloodstream infection in cancer patients treated with chemotherapy compared with non-cancer patients: a propensity score-matched analysis”, the authors reveal that PICC-related bloodstream infection rate was higher in cancer patients than that of non-cancer patients. Moreover, the authors also demonstrate that PICC-related bloodstream infections were frequently caused by Gram-negative bacteria in cancer patients; meanwhile, they were mainly caused by Gram-positive bacteria in non-cancer patients. The manuscript is well-written and clear, and the data are presented logically.

The authors thank the reviewer for their interest in our work.

Specific comment:

Interestingly, Gram-negative bacteria were frequently detected in PICC-related bloodstream infections developed in cancer patients. The authors are advised to briefly mention the microbial procedures to detect the causative pathogens in their hospital.

We thank the reviewer for their helpful observations. We have amended the method section with the microbiological procedures used in this study in accordance with the reviewer's comment. Please see method section, chapter 2.4.

Reviewer 3 Report

The manuscript, entitled “Peripherally inserted central venous catheter (PICC) related bloodstream infection in cancer patients treated with chemotherapy compared with non-cancer patients: a propensity score-matched analysis” described that PICC-related bloodstream infection rate was more than twice as high in cancer patients compared to non-cancer patients, and dual-lumen PICCs had a higher risk of PICC-related bloodstream infection than single-lumen PICCs. The study is meaningful to physicians to implement infection control measures in cancer patients, however, due to some drawbacks, my suggestion is major revision.

Major comments:

1. In this study, 240 were placed in cancer patients for chemotherapy and 481 in non-cancer patients. All these patients are collected from Nimes University Hospital from 01/04/2018, to 01/04/2019. Why did not the author collect patients from several other different hospitals? Whether procedures or treatment of physicians affect the results of the PICC-BSIs and affect the outcomes of this study?

2. In table 1, it shows that PICC duration (days) for cancer patients and non-cancer patients is 15,76 and 8.35, respectively, and p value is smaller than 0.001. It means the time of cancer patients and non-cancer patients carrying PICC is significantly different. The difference of this might lead to the difference of PICC related complications. Do authors agree with this opinion?

3. According to the results of this retrospective study, it is suggested to supplement some suggestion to the physicians on implement of PICC.

1. There are too many keywords. Normally, the number of keywords is no more than seven.

2. The introduction section does not show detailed and comprehensive background information to this study. Please expand it, for example, what is the difference of dual-lumen PICCs and single-lumen PICCs? Under what circumstances are they recommended?

3. In table 5, “/” before “year” is suggested to delete.

4. In table 4, “Cancer patients N = 240”, “Non-cancer patients N = 240”, the overall N should be 480, but it shows “Overall N = 420”. Please check it and make sure it is correct.

Author Response

The manuscript, entitled “Peripherally inserted central venous catheter (PICC) related bloodstream infection in cancer patients treated with chemotherapy compared with non-cancer patients: a propensity score-matched analysis” described that PICC-related bloodstream infection rate was more than twice as high in cancer patients compared to non-cancer patients, and dual-lumen PICCs had a higher risk of PICC-related bloodstream infection than single-lumen PICCs. The study is meaningful to physicians to implement infection control measures in cancer patients, however, due to some drawbacks, my suggestion is major revision.

The authors thank the reviewer for their interest in our work and for their thorough comments.

Major comments:

  1. In this study, 240 were placed in cancer patients for chemotherapy and 481 in non-cancer patients. All these patients are collected from Nimes University Hospital from 01/04/2018, to 01/04/2019. Why did not the author collect patients from several other different hospitals? Whether procedures or treatment of physicians affect the results of the PICC-BSIs and affect the outcomes of this study?

We fully agree with the reviewer that the study would have been more significant if data from other centers had been available. Please note that the present study is a secondary analysis of a retrospective, monocentric observational cohort. The study was single center, but it included a large number of patients from different oncology departments and a hematology department, so we think it represents a mix of different practices. Please see limit section.

  1. In table 1, it shows that PICC duration (days) for cancer patients and non-cancer patients is 15,76 and 8.35, respectively, and p value is smaller than 0.001. It means the time of cancer patients and non-cancer patients carrying PICC is significantly different. The difference of this might lead to the difference of PICC related complications. Do authors agree with this opinion?

As the reviewer correctly pointed out, the PICC indwelling time is an important parameter associated with PICC-related infections. We agree with the reviewer that cancer patients who have a higher PICC indwelling time than non-cancer patients are at greater risk of PICC-related infections. As a result, the rate of PICC-related bloodstream infections is higher in cancer patients (17%) than in non-cancer patients (2.9%), p<0.001. However, comparing incidences rather than rates allows the time factor to be considered (please see Method section, Statistical analysis, line 170, p4). The rate of PICC-related bloodstream infection is the number of PICC-related bloodstream infection for 100 patients exposed, whereas the incidence is the number of PICC-related bloodstream infection for 1000 catheter days. Thus, in our study we found that the incidence of PICC-related bloodstream infection is twofold higher in cancer patients than in non-cancer, please see table 2. The comparison by the Kaplan-Meier methodology confirm that the incidence of PICC-related bloodstream infection is significantly higher in cancer patients than in non-cancer (p < 0.007), please see figure 1. Our results therefore show that, for the same duration of PICC carriage, cancer patients are at greater risk of PICC-related bloodstream infections.

  1. According to the results of this retrospective study, it is suggested to supplement some suggestion to the physicians on implement of PICC.

As suggested by the review, we added in the discussion section a paragraph with the different suggestion to the physicians for improving care of cancer patients with PICC. See lines 305-322, p9.

Comments on the Quality of English Language

  1. There are too many keywords. Normally, the number of keywords is no more than seven.

Thanks to the reviewer for highlighting this mistake. The number of keywords is now seven.

  1. The introduction section does not show detailed and comprehensive background information to this study. Please expand it, for example, what is the difference of dual-lumen PICCs and single-lumen PICCs? Under what circumstances are they recommended?

As requested by the reviewer, the introduction has been expanded. Please see introduction section.

  1. In table 5, “/” before “year” is suggested to delete.

Done.

  1. In table 4, “Cancer patients N = 240”, “Non-cancer patients N = 240”, the overall N should be 480, but it shows “Overall N = 420”. Please check it and make sure it is correct.

Thanks to the reviewer for highlighting this mistake. “Overall N = 420” was replaced by “Overall N = 480”.

Round 2

Reviewer 3 Report

/

/